# RARβ Expression in Keratinocytes from Potentially Malignant Oral Lesions: The Functional Consequences of Re-Expression by De-Methylating Agents

**DOI:** 10.3390/cancers13164064

**Published:** 2021-08-12

**Authors:** Raghu Radhakrishnan, Hannah L. Crane, Marc Daigneault, Kanaka Sai Ram Padam, Keith D. Hunter

**Affiliations:** 1Oral Pathology, Manipal College of Dental Sciences, Manipal Academy of Higher Education, Manipal 576104, India; raghu.ar@manipal.edu; 2Academic Unit of Oral and Maxillofacial Medicine and Pathology, School of Clinical Dentistry, University of Sheffield, Sheffield S10 2TN, UK; h.crane@sheffield.ac.uk; 3Medicines Discovery Catapult, Mereside, Alderley Park, Alderley Edge, Cheshire SK10 4TG, UK; marc.daigneault@md.catapult.org.uk; 4Cell and Molecular Biology, Manipal School of Life Sciences, Manipal Academy of Higher Education, Manipal 576104, India; kanaka.sairam@learner.manipal.edu; 5Departmentt of Oral Pathology and Oral Biology, Faculty of Dentistry, University of Pretoria, Pretoria 0002, South Africa

**Keywords:** RARβ, oral cancer, oral potentially malignant lesions, PMOL, retinoids, chemoprevention

## Abstract

**Simple Summary:**

Patients may develop white or red patches of the lining of the mouth with an increased risk of developing oral cancer. Treatment with Vitamin A derivatives (retinoids) results in some improvement in these lesions, but this is not maintained, and there are side effects. We know that the cells of the mouth lose cellular receptors for retinoids as these lesions develop, initially by a reversible alteration to the DNA (DNA methylation). Drugs, such as 5-AZA-CdR, which reduce DNA methylation, may restore sensitivity to the effects of retinoids. Treatment of a panel of cells from mouth precancer white patches with retinoids, 5-AZA-CdR and a combination results in varied responses: some cells re-sensitise to retinoids, whereas in others, the main effects on cell division rate and cell lifespan seem related to the effects of 5-AZA-CdR alone. These findings help us to understand the varied responses to retinoids in the clinical setting.

**Abstract:**

Loss of RARβ2 expression by promoter methylation is an early event in oral carcinogenesis. Understanding the mechanisms and consequences of RARβ loss may aid in understanding the disappointing results of retinoid chemoprevention trials. This study aimed to describe the effects of all-trans retinoic acid (ATRA) and the de-methylating agent 5-Aza-2′ deoxycytidine (5-AZA-CdR) on a panel of immortal potentially malignant oral lesion (PMOL) cell cultures. RARβ expression was assessed in PMOL tissues by immunohistochemistry. Cells were treated with ATRA ± 5-AZA-CdR, and the effects on the cell cycle and senescence were assessed. In PMOL tissues, RARβ expression was variable, but lower in biopsies which gave rise to immortal cell cultures. Treatment of iPMOL cells with ATRA resulted in little change in RARβ expression, but the addition of 5-AZA-CdR resulted in significant increases. The effects on the cell cycle and senescence were variable and may be related to 5-AZA-CdR, as this has wider effects on the cell cycle. Overall, the response of iPMOL cells to ATRA and 5-AZA-CdR treatment was variable and is dependent on several factors, including RARβ-promoter methylation. These findings may help to explain the lack of consistent effect of retinoids in PMOLs seen in chemoprevention trials.

## 1. Introduction

Potentially malignant oral lesions (PMOLs) present variably in the mouth, including white or red lesions (leukoplakia and erythroplakia). The risk of the development of oral squamous cell carcinoma (OSCC) in such lesions has been related to several factors, including patient age, the size, clinical appearance and site of the lesion, and degree of epithelial dysplasia [1,2]. There is, however, a lack of validated biomarkers for the prediction of malignant transformation [3]. Given the well-established concept of field cancerisation of the oral mucosa, it has been demonstrated that the underlying molecular lesion within the oral mucosa may be significantly more extensive than either the clinical or histological abnormality [4]. The extent of this genetic abnormality means that it may not be possible, or even desirable, to eradicate this by surgical means, even if the clinically evident lesion can be removed. Thus, there has been much interest in (chemo)prevention of progression to OSCC over many years.

Much of the interest in chemopreventative agents has focussed on Vitamin A and related compounds, often referred to collectively as retinoids. Initial studies used topical Vitamin A, but subsequent clinical investigations used a number of naturally occurring or synthetic analogues, including 13-cis-retinoic acid [5,6,7]. Several moderately sized clinical trials have been completed using these agents. Whilst initially promising, with reductions in the size of clinical lesions, the overall long-term outcomes in terms of cancer prevention were variable [6,8,9,10]. Additionally, some authors reported significant side effects and relapse of lesions after cessation of treatment [11]. In summarising this literature, the Cochrane review of interventions in oral leukoplakia concluded that there is currently insufficient high-quality evidence for any intervention which will reduce the malignant transformation rate in OPMLs [12]. This has resulted in less interest in the literature on the effects of retinoids and other vitamin A analogues for OSCC prevention in recent years. 

The effects of retinoids on oral mucosa are complex. In general, treatment of oral epithelial cells with retinoids results in inhibition of cell proliferation and terminal differentiation [13]. The expression of certain retinoic acid receptors in normal oral keratinocytes depends on retinoid treatment. In some mortal cell cultures derived from PMOLs, constitutive expression of the retinoid receptor Retinoic Acid Receptor Beta (NR1B2, HGNC: 9865, hereafter referred to as RARβ) has been described [14]. However, in the progression of PMOLs to OSCC, loss of certain retinoid receptors has been described in both cell/tissue types, most notably the RARβ isoform RARβ2 [14,15]. There are four isoforms of RARβ which have been generated by alternative splicing (www.uniprot.org/uniprot/P1082, accessed on 5 August 2021). Loss of RARβ2 has been associated with a number of other well-established alterations in oral epithelial cells on progression to OSCC via PMOLs, namely, loss of p16^INK4a^ expression, p53 mutations and activation of telomerase [16]. This combination of alterations has been associated with the bypass of replicative senescence, allowing cells to become immortal. The basis of loss of RARβ2 expression, which may occur at the PMOL stage, is hypermethylation of the RARβ2 promoter [17]. The role of other associated molecules which are important in the modulation of retinoid receptor signalling, such as the cellular retinoid-binding proteins (cRPBs) [18], in this context is not known.

Various approaches have been reported to address these issues associated with retinoid therapy of PMOLs, including the use of newer synthetic retinoid analogues (for example, fenretinide: [9] and studies focussed on the identification of biomarkers, which may predict retinoid sensitivity [9,19,20,21,22]. A further possible approach has been used in other cancers: the addition of de-methylating agents to retinoids treatment. These agents, including 5-aza-2′-deoxycytidine (5-AZA-CdR), have been used clinically in the treatment of myelodysplastic syndromes and AML, where retinoid therapy is also employed. The effects of such combinations have been reported in vitro and in vivo clinical studies, with promising effects [23,24]. The approach has also been assessed in solid tumours, such as breast carcinoma and neuroblastoma [25,26]. Proof of principle has already been established in PMOL and OSCC cells [16,17], but an in-depth assessment of the functional effects in PMOLs has not been reported.

This raises the question of whether such an approach would be feasible in oral epithelial cells derived from PMOLs, laying the basis for possible clinical studies of such an approach in the chemoprevention of OSCC. In the present study, we aim to confirm the de novo methylation of RARβ in oral epithelial dysplasia and understand its role in cellular immortalisation and abrogation of cellular senescence. We test the hypothesis that administration of 5-AZA-CdR alone and/or in combination with All trans-retinoic acid (ATRA) leads to re-expression of RARβ, reversal of immortalisation, and reinduction of the senescence programme in a panel of immortalised primary cultures.

## 2. Materials and Methods

### 2.1. Cell Lines and Culture Conditions

The experimental work described used a unique cohort of cell cultures derived from a variety of potentially malignant oral lesions (PMOL), all part of the Beatson Institute for Cancer Research cell culture collection. These have varied proliferative lifespans, with some undergoing replicative senescence (D6 and D30), whilst others are immortal (D19, D20, D34 and D38; Table 1). The p16−/TERT+ immortalised NOK culture FNB6^TERT^ was used as a control. The molecular features that define these cultures have been previously described [16,27]. The cultures were maintained on an irradiated Swiss 3T3 feeder layer in Green’s medium at 37 °C with 95% humidity and 5% CO_2_. The medium consisted of a 1:3 (*v*/*v*) mixture of Dulbecco’s modified Eagle’s medium (DMEM) supplemented with 4500 mg/L glucose GlutaMAX™ I and sodium pyruvate (Gibco, Paisley, UK) and Ham’s F12 medium supplemented with L-glutamine and sodium bicarbonate (Biosera, East Sussex, UK) with heat-inactivated fetal calf serum (Biosera, East Sussex, UK). This was supplemented with adenine (0.025 µg/mL) insulin (5 µg/mL), 3, 3, 5-Tri-iodothyronine/Apo-Transferrin (1.36 ng/mL T3 and 5µg/mL apo-transferrin), hydrocortisone (4 µg/mL), epidermal growth factor (5 ng/mL), amphotericin B (0.625 µg/mL), cholera toxin (8.47 ng/mL), penicillin (100 IU/mL), and streptomycin (100 µg/mL). The irradiated 3T3 feeder layer was removed by treatment with 0.05% trypsin/0.02% EDTA (*w*/*v*) (Sigma Aldrich, Dorset, UK) before keratinocyte trypsinisation. 

### 2.2. ATRA and 5-Aza-CdR Treatment of Cells

All-trans-retinoic acid (Sigma-Aldrich, Dorset, UK), dissolved in dimethyl sulfoxide (DMSO) at a concentration of 10^−2^ M, and stored in dark at −70 °C, was diluted in growth medium to a final concentration of 10^−6^ M immediately before each experiment. Control cultures received the same amount of DMSO as treated cultures. 5-aza-2′-deoxycytidine (5-Aza-CdR) (Sigma, St. Louis, MO, USA) was dissolved in DMSO at a concentration of 2 × 10^−3^ M, stored in small aliquots the dark at −70 °C. Cell lines were treated with 0.5 µM of 5-Aza-CdR with media refreshed every 24 h for 5 consecutive days before genomic DNA and total RNA were extracted and tested for methylation status as well as restoration of RARβ expression. RARβ2 induction was analysed in the cells treated using 1, 5, or 10 µM of ATRA for 3 days. Untreated cells were used as an experimental control, and cells treated with a working concentration of DMSO were used as solvent control. 

### 2.3. cDNA Synthesis and RT-qPCR

Total RNA from the cultured cell lines was isolated using RNeasy Mini Kit (Qiagen, Manchester, UK) and treated with DNase I (Qiagen, Valencia, CA, USA). The quantity and quality of DNA and RNA were analysed by 1000 NanoDrop Spectrophotometer V3.7 (Thermo Scientific, Hemel Hempstead, UK). An input amount of 2 μg was used for Reverse transcription PCR, and cDNA synthesis was performed using a High Capacity cDNA Reverse Transcription kit (Applied Biosystems, Foster, CA, USA) following manufacturer’s instructions in the DNA Engine Dyad thermal cycler (Bio-Rad, Hercules, CA, USA). 

Quantitative real-time PCR (qPCR) was performed using ABI Prism 7900 Fast Sequence Detection Instrument (Applied Biosystems, Foster, CA, USA) for the relative gene expression of RARβ2 (Hs00977141_mH), CDKN2A, (Hs99999189_m1), CDKN1A (Hs00355782_m1), IVL (Hs00846307_s1), ITGB1 (Hs05351551_g1), and CRBP1 (Hs01011512_g1), with β-2-microglobulin (B2M) (Assay ID: Hs99999907_m1) as a housekeeping gene using Taqman Assays procured from Thermo Fisher Scientific. All experiments were performed in triplicate, analysis was carried out using 2^^−ΔCt^ method [28] to calculate the mRNA expression relative to the B2M, and data was represented as mean with standard error. 

### 2.4. Western Blotting

Following trypsinisation, cells (D6, D30, D34, D38, D19, D20, FNB6) were washed in 1× PBS (pH 7.0) and lysed in RIPA buffer (Sigma Aldrich, Poole, UK) containing protease and phosphatase inhibitors (Roche, West Sussex, UK). Protein preparation was carried out by centrifugation of cells at 12,000 rpm, 4 °C for 20 min, following which the supernatant was aspirated. The total protein concentration was determined using Pierce BCA Protein Assay Kit as per the manufacturer’s protocol (Thermo Scientific, Hemel Hempstead, UK). A total of 20 μg of total protein was loaded onto 4–12% polyacrylamide precast gels (NuPAGE™ 4 to 12%, Bis-Tris, 1.0 mm, Mini Protein Gel, Novex, Thermo Scientific, Hemel Hempstead, UK). After transferring the gel onto nitrocellulose membrane using an iBlot Dry Blotting System (Life Technologies, Carlsbad, CA, USA) for 7 min, the membranes were washed with Tris buffer and blocked with 5% dried milk in Tris-buffered saline (TBS) containing 0.05% Tween-20, for 1 h and incubated overnight at 4 °C with the following primary monoclonal antibodies: (Anti-RARβ antibody (EPR2017; (ab124701)) at 1/1000 dilution; Anti-p21 antibody (EPR18021; (ab188224)) at 1/1000 dilution; Anti-CDKN2A/p16INK4a antibody (DCS50.1; (ab16123)) at 1/5000 dilution and incubated for 1 h with primary anti-β-actin antibody (1:5000), Sigma Aldrich, Poole, UK). Membranes were then incubated in horseradish peroxidase (HRPO) conjugated with anti-rabbit IgG (1:3000, ab6721, Abcam (Cambridge, UK)) for 1 h and developed with SuperSignal West Pico chemiluminescent substrate (Thermo Scientific, Hemel Hempstead, UK). HeLA and irradiated 3T3 cells’ lysate were used as experimental controls. 

### 2.5. Growth Inhibition Assay and Flow Cytometry

Cells were seeded at a density of 1 × 10^5^ cells/well in a 6-well plate as required. Post 24 h, the cells were treated with different concentrations of ATRA, 0.5 µM 5-Aza-CdR and a combination of 0.5 µM 5-Aza-CdR and 1 µM ATRA for 3 days in Green’s medium containing 10% FCS. The cells after the treatment were labelled with EdU (5-ethynyl-2′-deoxyuridine) at a concentration of 10 μM for 1–2 h (Click-iT^®^ EdU Alexa Fluor™ 488 Assay Kits for Flow Cytometry, Life technologies; excitation 488 nm/emission 530 nm). EdU is a nucleoside analogue to thymidine and is incorporated into DNA during active DNA synthesis. Following EdU labelling, the cells were washed once with 3 mL of 1% BSA in PBS, pelleted, and the supernatant removed. The cell pellet was dislodged, and 100 μL of Click-iT^®^ fixative was added and mixed well. Cells were incubated for 15 min in the dark, then washed once with 3 mL of 1% BSA in PBS, pelleted, and the supernatant removed. Cells were resuspended in 100 μL of 1× Click-iT^®^ saponin-based permeabilisation and wash reagent and mixed well and incubated. Appropriate amounts of Click-iT^®^ Plus reaction cocktail was used within 15 min of preparation, and 0.5 mL of Click-iT^®^ Plus reaction cocktail was added for each tube, mixed well, and incubated for 30 min and protected from light. Following this, the cells were washed once with 3 mL of 1× Click-iT^®^ saponin-based permeabilisation and wash reagent and resuspended in 500 μL of 1× Click-iT^®^ saponin-based permeabilisation and wash reagent. A total of 50 µL of RNase (100 µg/mL solution) was added and incubated for 15 min at 37 °C. Finally, the nucleic acid dye, TO-PRO-3 (Thermo Fisher Scientific, excitation 643 nm/emission 661 nm), was added at a concentration of 0.1 µM and incubated for 15 min in the dark. Samples were analysed by flow cytometry using an LSRII flow cytometer (Becton Dickinson, Oxford, UK) and FACSDiva™ versio 7 acquisition software (Becton Dickinson, Oxford, UK). Cells were gated based on forward (FSC) and side (SSC) light scatter, and 10,000 events per sample were recorded. Data were analysed using FlowJo analysis software v7.6.5 (Tree star Inc, Ashland, OR, USA). Apoptotic cells in the sub-G0 region were excluded, and standard flow cytometry methods were used for determining the percentage of G1/S/G2-phase cells in the population, as previously described [29].

### 2.6. Promoter Methylation Analysis

Promoter methylation analysis was performed by MethylScreen technology [30], using EpiTect Methyl II PCR Assay kit, Qiagen. Briefly, genomic DNA was extracted and purified from the treated and untreated PMOL cell lines using QIAmp DNA mini kit and aliquoted in equal amounts of DNA (250 ng) into four tubes labelled mock (M0), methylation-sensitive (Ms), methylation-dependent (Md), and methylation-sensitive-dependent (Msd) restriction enzymes. All four cocktail reaction tubes were incubated at 37 °C for 6 h and then at 65 °C for 20 min using DNA Engine Dyad thermal cycler (Bio-Rad, Hercules, CA, USA). Quantitative PCR for determination of methylation status was performed in ABI Prism 7900 Fast Sequence Detection Instrument (Applied Biosystems, Foster, CA, USA) using 5 μL of remaining input genomic DNA post the restriction digestion with qPCR master mix (RT2 qPCR SYBR Green/ROX Master Mix, Qiagen, Manchester, UK: Cat number 330520), and were dispensed into a PCR array plate containing pre-aliquoted primer mixes (EpiTect Methyl II qPCR Primer Assay) according to manufacturer instructions. Thermal cycling conditions used were: 95 °C for 10 min (1 cycle), then 99 °C for 30 s and 72 °C for 1 min (3 cycles), and finally, 97 °C for 15 s and 72 °C for 1 min (40 cycles). The raw Ct values obtained post the run were pasted into EpiTect Methyl II PCR Array Microsoft Excel-based data analysis template (supplied), which automatically calculates the relative amount of methylated and unmethylated DNA fractions.

### 2.7. Methylation-Specific PCR (MSP) and Bisulfite PCR for Restriction Analysis 

The promoter methylation status of RARβ was determined by MSP. Briefly, 2 μg of genomic DNA was treated with sodium bisulfite for 16 h. Following this, DNA was desulfonated and purified using the Wizard DNA Clean-up system (Promega, Madison, WI, USA). An input quantity of 2 μL of converted DNA was used as a template for performing MSP with two sets of primers to differentiate between methylated and unmethylated regions as described [31]. The primer sequences specific for methylated sodium bisulfite DNA (FP: 5′-TCGAGAACGCGAGCGATTCG-3′ (sense) and RP: 5′-GACCAATCCAACCGAAACGA-3′ (anti-sense); 146 bp, T_H_ −58 °C) and those for unmethylated sodium bisulfite DNA (FP: 5′-TTGAGAATGTGAGTGATTTGA-3′ (sense) and RP: 5′-AACCAATCCAACCAAAACAA-3′; 146 bp, T_H_ −50 °C) were used [32]. 

Furthermore, methylation of the RARβ-promoter region was determined using combined bisulfite PCR followed by restriction analysis as described earlier (COBRA) [33]. The primer sequences used for amplifying the modified DNA were forward: 5′-AAGTAGTAGGAAGTGAGTTGTTTAGA-3′ and reverse: 5′-CCAAATTCTCCTTCCAAATAA-3′ yields an amplicon of 207 bp product as published [17]. The amplified products were then digested with TaiI for RARβ2 (MBI Fermentas, Hanover, Germany) and subjected to electrophoresis on 3% agarose gel and visualised using ethidium bromide. Human methylated lymphocyte DNA generated in vitro by CpG methyltransferase (*Sss*I; New England Biolabs, Inc., Beverly, MA, USA) was used as a positive control and the non-template (negative) control was included as in every PCR run.

### 2.8. DNA Cloning and Sequencing

Primers employed to generate the target region of interest for cloning were the same used for COBRA to clone DNA fragments into a Pcr2.1 TOPO vector (Invitrogen Corporation, Carlsbad, CA, USA) according to manufacturer’s instructions. The generated vector construct was then transformed into chemically competent *E. coli*, and positive colonies were selected based on blue–white screening for propagation. Plasmid DNA was extracted and purified using Qiagen Plasmid Mini Kit (Qiagen, Valencia, CA, USA) and sequenced using ABI PRISM 377 DNA sequencer (Applied Biosystems, Foster, CA, USA) at the DNA-sequencing core facility at the University of Sheffield, UK.

### 2.9. Immunofluorescence 

Cells were plated in eight-well glass chamber slides (Thermo Scientific™ Nunc™ Lab-Tek™ Chamber Slide System) prior to processing. The 1 × 10^4^ cells (D19, D20, D34, D38 and FNB6) were seeded onto coverslips and cultured until they reached 95% confluence. Cells were fixed in 4% paraformaldehyde for 15 min and permeabilised using Triton-X-100 (Sigma, Poole, UK) for 15 min. Blocking of non-specific binding was carried out using unlabelled serum from the same species as the second antibody for 1 h at room temperature. Samples were incubated with the primary antibody Anti-HIRA/HIR antibody (EPR7416) (ab129169) at 1:200 dilution at 4 °C overnight. Goat anti-rabbit IgG (Alexa Fluor^®^ 488, ab150077) was used as the secondary antibody at 1:1000 (2 µg/mL) dilution for 1 h at room temperature (RT). Cells were counterstained with DAPI (Invitrogen) for 10 min at RT in the dark. 1× PBS with 0.05% Tween-20 was used throughout the protocol for adequate washing. Coverslips were mounted on microscopic slides with antifade solution (VectaShield, Vector Laboratories, Burlingame, CA, USA). Cells incubated with IgG and secondary antibodies served as a background control. Stained cells were examined using a Leica DMRB fluorescence microscope and ×40/0.7 objective lens (Leica Microsystems, Wetzlar, Germany); for these experiments, image processing was performed by LAS AF software (Leica Microsystems). Three or more foci per nucleus were considered to be positive for HIRA.

### 2.10. Senescence β-Galactosidase Staining

Senescent keratinocytes were identified using Senescence β-Galactosidase Staining Kit (Cell Signalling Technology, Inc., Beverly, MA, USA). β-Galactosidase Staining Solution at final pH of 6.0 stain positively for senescent cell, as described by [34]. Briefly, cells were grown on a 35 mm Petri dish, washed once with PBS, fixed with PBS containing 2% formaldehyde and 0.2% glutaraldehyde, then washed in PBS supplemented with 1 mM MgCl_2_. Cells were stained in senescence-associated β-galactosidase staining solution (150 mM NaCl, 2 mM MgCl_2_, 5 mM potassium ferricyanide, 40 mM citric acid, and 12 mM sodium biphosphate, at pH 6.0. containing 1 mg/mL of 5-Bromo-4-chloro-3-indolyl β-d-galactoside overnight at 37 °C in a dry incubator without CO_2_. The plates were sealed with parafilm to prevent evaporation. For long-term storage of the plates, staining solution was removed, and the cells were overlayed with 70% glycerol and stored at 4 °C. To ensure a representative count, each culture was divided into quarters, and at least two fields were photographed with an Olympus IMT-2 phase-contrast microscope. A minimum of 500 cells was counted on each occasion. The percentage of senescent cells present was defined as the percentage of senescence-associated β-galactosidase positive cells out of the total number of cells present. Average percentages and SD were calculated from three independent experiments. 

### 2.11. Tissue-Engineered Oral Mucosa

De-epidermised acellular dermis was incubated in fresh DMEM for 48 h at 37 °C to confirm sterility. Processed DED was cut into 2 cm × 2 cm squares and placed into 6-well plates submerged in Green’s media. Chamfered surgical stainless-steel rings with an internal diameter of 8 mm were pushed onto the DED to provide a liquid-tight seal. For the tissue-engineered dysplastic oral mucosa models (TEDOM), 2.5 × 10^5^ D6, D30, D34, D38, D19 and D20 cells and 5 × 10^5^ normal oral fibroblasts (NOFs) was used. Medium within the ring was replaced after 24 and 48 h. After 72 h, the ring was removed, and the composites were placed onto stainless steel grids with medium added to the underside of the composite model to allow culture at the air-to-liquid interface. For all models, the medium was changed 2–3 times/week, and the composites were fixed at day 14 for TEDOM in 10% buffered formalin for 48 h [35].

### 2.12. Patient Tissue Samples and Immunohistochemistry

Two cohorts of patient samples were analysed by immunohistochemistry: one whose samples directly matched the cell cultures used in the main part of the project (*n* = 6) and a separate, unrelated cohort of oral premalignant lesions (*n* = 10). Oral precancerous tissues from surgically excised oral dysplasia lesions representing different grades of dysplasia and adjacent normal mucosal tissue from the archive of the Department of Oral and Maxillofacial Pathology, University of Sheffield. RARβ expression was assessed by immunohistochemistry in a cohort of dysplastic lesions of various grades with a particular emphasis on tissue heterogeneity. All the original histological diagnoses were reviewed by two independent examiners (RR and KDH). Then, 4 μm sections were cut from FFPE blocks and mounted on APES coated slides. Deparaffinization and hydration of the tissue sections were carried out through two changes of xylene and graded alcohols. Heat-induced antigenic epitope retrieval was carried out in Sodium citrate buffer (10 mM Sodium Citrate, 0.05% Tween-20, pH-6.0). After washing sections twice in TBS +0.025% Triton-X-100 with gentle agitation, sections were blocked in 10% normal serum with 1% BSA in TBS for 2 h at room temperature. After draining slides, an anti-RARβ antibody (EPR2017) (ab124701) was added at 1:100 dilution in TBS with 1% BSA and incubated overnight at 4 °C. The following day, sections were rinsed twice in TBS with 0.025% Triton-X-100 with gentle agitation. Sections were then incubated in a biotinylated secondary antibody (Vectastain Elite ABC Kit) for 30 min. Following a 2 × 5 min wash in TBS, sections were incubated with VECTASTAIN Elite ABC reagent for 30 min, washed, and visualisation was developed with 3,3′-Diaminobenzidine (DAB). Eventually, sections were counterstained, dehydrated, cleared, and mounted. 

Each spot image was submitted to colour deconvolution to separate the blue colour from haematoxylin and the brown colour from DAB using the plugin in ImageJ software (National Institute of Health, Bethesda, MD, USA). The positive labelling (brown colour) was selected using the threshold tool of ImageJ (from 0 to 127 brown tones). First, the image was processed by colour deconvolution using the two vectors, hematoxylin and DAB. Assessment of intensity was not carried out as the immunostains are not stoichiometric. Then, the processed image was adjusted for optimal threshold. The upper and the lower limit for both the DAB alone and hematoxylin was adjusted and the particles in both were separately analysed. The final score was calculated as ((positive labelling area/tumour area) × 100)). A qualitative interpretation of immunohistochemical signal was performed, blinded to diagnosis and clinical data. 

### 2.13. Statistical Analysis

The data analysed was represented as mean with standard error derived from at least three independent experiments. Comparisons among the groups were performed using one-way ANOVA followed by post hoc Tukey’s test or Dunnett’s test. Comparison between the two groups was performed using paired or unpaired *t*-tests. All the statistical analysis mentioned was performed using GraphPad prism statistical software v8.0 (GraphPad Software Inc., San Diego, CA, USA). A *p*-value of <0.05 was considered statistically significant. 

## 3. Results

### 3.1. RARβ2 Expression Is Lost in iPMOL Cultures and Reduced in Their Matched Tissues, but Expression Varies in All PMOL Tissues 

RARβ2 is expressed by mortal PMOL cells (D6 and D30), whereas expression has been lost in immortal OPML (iPMOL) cell cultures (D19, D20, D34 and D38) and the immortalised NOK culture FNB6 (Figure 1A), as assessed by qPCR. Total RARβ protein expression was reduced in iPMOL cells (Figure 1B). In keeping with earlier reported patterns of expression in these cells, there is lower p16 and p21 expression in the immortal cells [14,16,27]. Markers of differentiation (IVL and ITB1) and expression of cRBP1 are also reduced in iPMOL cells (Figure 1A,B). 

The extent of RARβ expression in the biopsy samples matched to the cell cultures shows that RARβ expression was lower in the tissues that gave rise to immortal cell cultures (Figure 1C; Table 1). This indicates that the loss of RARβ expression is not merely an in vitro phenomenon. Nevertheless, all tissues did express RARβ to some extent, indicative of a mixed cell population in these PMOL tissues, as we have previously suggested [36]. Expression of RARβ in an extended clinical cohort (*n* = 10) confirmed this heterogeneity in the expression of RARβ2 (Figure 1D). The proportion of cells expressing RARβ2 was variable and was not directly related to the grade of epithelial dysplasia (Table 2). The expression of cRBP1 in this biopsy cohort was very variable. RARβ staining in the 3D tissue-engineered oral mucosa (TEM) models developed from iPMOLs showed a heterogeneous staining pattern, but which was higher overall than in the parent tissue biopsies (Figure 1E). 

### 3.2. The RARβ-Promoter Was Hypermethylated in 3/4 Immortal Oral Dysplastic Cell Lines

Analysis of DNA methylation of the RARβ-promoter region was performed using Epitect Methyl II qPCR assay specific for RARβ-promoter CpG island (GenBank: X56849.1). The RARβ promoter was hypermethylated in D19 and D34 cells, with the methylated fraction of DNA noted to be 71.12% and 51.91%, respectively. A significant reduction in methylation was noted after treatment with 5-AZA-CdR (*p* < 0.001). The methylation level in the D20 cells was 11.76%, which was lower than seen in D19 and D34. However, significant de-methylation was noted upon treatment with 5-AZA-CdR (*p* < 0.0017). In D38, the RARβ promoter was unmethylated and showed no change with treatment (Figure 2A). 

To eliminate the potential false positives and to confirm the extent of methylation, methylation-specific PCR was carried out. Methylation-specific PCR of bisulfite-treated DNA from four IMDs showed a 146 bp transcript using primers specific for methylated (M) and unmethylated (U) DNA in the untreated (−) and treated (+) cells. A reduction in the band intensity in D19 and D34 and a modest reduction in D20 were noticed when treated with 2 μM 5-AZA-CdR. The D38 cells remained unchanged in both the methylated (M) and unmethylated (U) lanes before and after treatment (Figure 2B). 

Further validation of DNA methylation and de-methylation by the action of 5-AZA-CdR was confirmed by bisulfite sequencing. This was followed by COBRA, where a 207 bp of bisulfite modified DNA of the promoter was amplified, and the PCR product was digested using the TaiI restriction enzyme, which specifically cuts at ACGT^ sites generating upper unmethylated and a lower methylated DNA fragment, whereas unmethylated DNA remained uncut (Figure 2C,D). Analysis of the 11 CG sites in all the iPMOL cells was determined by sequencing followed by comparing the peak height of cytosine with the sum of the height of cytosine and thymidine signals (Figure 2E). 

### 3.3. Treatment with 5-Aza-CdR and ATRA Variably Alters Expression of RARβ, p16, and p21 in iPMOL Cells

The extent of expression of RARβ in D19, D20, D34 and D38 was assessed after treatment of the cells with various concentrations of ATRA and 5-AZA-CdR separately and in combination. In D19 and D34, RARβ expression was highest (*p* < 0.0001) when treated with a combination of 0.5 μM 5-AZA-CdR and 1 μM ATRA. A concomitant increase in CDKN1A (*p* < 0.0001) and CDKN2A (*p* < 0.002) expression was noted when the cells were treated with ATRA and 5-AZA-CdR alone and/or in combinations (Figure 3A,B). D20 showed a re-expression of RARβ (*p* < 0.0001) when treated with 5-AZA-CdR and ATRA combination, and CDKN1A re-expression was significant on treatment with 10 μM ATRA (*p* = 0.0001), but changes in CDKN2A expression were not statistically significant (Figure 3A,B). In D38, RARβ was re-expressed on ATRA treatment (*p* < 0.03). CDKN1A was expressed on treatment with ATRA in combination with 5-AZA-CdR (*p* < 0.0001), whereas there was no significant difference in CDKN2A expression (Figure 3A,B). Although the combination of 5-AZA-CdR and ATRA produced the highest levels of RARβ in iPMOL cells, ATRA or 5-AZA-CdR alone when used in varying concentrations induced RARβ re-expression (*p* < 0.05). These results imply that the response is cell-line-specific and dose-dependent.

In 3D TEM models, RARβ staining of the models showed a significant increase in expression after treatment with ATRA (Figure 3C and Table 3). Under 5-AZA-CdR treatment (alone or in combination with ATRA), the 3D TEM models were insufficiently robust for further assessment (included for completeness in Figure 3C). 

We also identified changes in the expression of cRBP1 in these cultures (Figure 1A and Appendix A). Treatment with 5-Aza-CdR and 5-Aza-CdR + ATRA resulted in increased expression in D34 and D20, with little effect in D19 and D38 (Appendix A). The mechanism of this is unclear as the cRBP1 promoter did not show CpG island methylation in any case.

### 3.4. Treatment with 5-Aza-CdR and ATRA Increases the Proportion of Senescent iPMOL Cells

The HIRA foci assay showed an increased accumulation of senescence-associated heterochromatin foci (SAHF) in iPMOL cells when treated with the combination of 5-AZA-CdR with ATRA (*p* < 0.0007: Figure 4A). D34 and D38 cells also showed a significant increase in HIRA foci when treated with varying concentrations of ATRA (*p* < 0.0001) or 5-AZA-CdR (*p* < 0.002) (Figure 4A). 

The mean percentage values of senescence-associated β-galactosidase (Saβ-gal) activity increased upon treatment with ATRA and 5-AZA-CdR in D20, D34, and D38 (*p* < 0.0001) cells, and less so in D19, when compared to untreated cells (Figure 4B). However, D38 cells showed relatively uniform SA-β-gal positivity across the concentrations tested with or without combination, with only a small but significant increase (*p* < 0.0002). As with the HIRA foci assay, D34 and D38 cells showed a significant increase in SA-β-gal activity when treated with the various concentrations of ATRA or 5-AZA-CdR when compared to untreated cells (*p* < 0.0002: Figure 4B). 

### 3.5. ATRA and 5-AZA-CdR-Treated IMDs Show G2/M Arrest and Increased Sub-G0 Phase

Treatment of the iPMOL cells with 5-AZA-CdR and 5-AZA-CdR + ATRA resulted in accumulation of cells in the G2 phase of the cell cycle in D19, D20, and D34, with little effect seen in D38 (Figure 5A–E) when compared to vehicle control (DMSO). The effect appeared to be primarily mediated by 5-AZA-CdR, as ATRA alone had little effect nor further increased the G2 accumulation seen when used in combination with 5-AZA-CdR. Treatment with ATRA between 5–10 µM increased the proportion of cells in G1 for D19 only (Figure 5D), with a corresponding decrease of newly synthesised DNA. Apoptosis was assessed by quantifying sub-G0 content, which typically represents fragmented DNA. A small increase in sub-G0 events was seen with 5-AZA-CdR treatment of D19 cells (2.92% vs. 0.73% vehicle control: Figure 5C); this is unlikely to account for the significant reduction in S-phase DNA.

## 4. Discussion

The potential of the combination of effects of a methylating agent, such as 5-AZA-CdR with retinoids, has been explored in a number of different malignant and pre-neoplastic conditions [23,24,25]. The observation of loss of expression of some retinoid receptors by promoter methylation, associated with loss of retinoid sensitivity and thus retinoid-related modulation of differentiation and cell cycle arrest, indicates that this is potentially a useful concept for chemoprevention. However, cell culture data in many cases have been conducted on a very limited panel of cells, and, in some publications, only one. This leaves several open questions as to the variability of response which may be associated with the biological variability seen in the development of cancer and pre-neoplastic conditions. Proof of principle has already been established in PMOL and OSCC [16,17], but an in-depth assessment of the variability of response at the level of the effects on the cell cycle and replicative potential has not been reported. 

Whilst the loss of RARβ expression has been reported in OSCC tissues, this has not been explored in PMOL tissues. Previous work, confirmed and extended in this report, has demonstrated that RARβ2 expression is lost in most immortal oral precancer cells (Figure 1A: [14]). This pattern of expression is reflected, to some extent, in the original tissues, but tissue expression of RARβ varies. The tissues used for this assessment are those from the original diagnostic biopsy, from which half of the sample was used to derive the cell culture. The IHC staining pattern demonstrates that dysplastic oral mucosa consists of a mixed cell population of RARβ expressing and RARβ negative cells. Whilst it has not been possible to directly correlate this with other elements of the immortal phenotype in these tissues (p53 mutations, p16 loss, Telomerase activation), other studies have reported heterogeneity in PMOL oral mucosa, for example, the appearance of “patches” of p53 mutation, indictive of heterogeneous cell populations [37]. Given that some of the PMOL tissues gave rise to cell cultures that can senesce, it is likely that these lesions contain populations of cells with a mixed proliferative potential, and indeed, some cells may be senescent or at least retain the potential to senesce. The retention of these cells within PMOL mucosa may give the lesion a variable ability to respond to retinoids, as seen in the numerous clinical studies reported over many years [6,8,9,10]. The expression of RARβ in the TEM models is higher than in the original parent tissues (Figure 1E), which is expected when using a single cell population for the generation of these models: however, there is variability in expression, and an increase in expression is seen on ATRA treatment (Table 3). This may indicate complex effects of differentiation in the 3D model which are not seen in monolayer culture and which will require further investigation. 

Previous studies of retinoid receptor expression in PMOL tissues are very limited and hampered by a lack of sufficiently specific antibodies: indeed, the antibodies used in this study have been challenging to use. They are not specific for RARβ2 and will pick up other RARβ isoforms, which may explain the discrepancies between the qPCR and WB data. The overall expression of RARs and RXRs appears to be increased in PMOL and SCC, indicating complex changes in the spectrum of retinoid receptors in OSCC development [38,39], but protein expression of subtypes beyond this has not been reported. Studies using in situ hybridisation for RAR and RXR mRNA demonstrate loss of RARβ expression in about 50% of lesions with a further reduction in OSCC, keeping with our findings [40]. 

RARβ2 is re-expressed following administration of 5′AzaC to a variable extent: markedly in 2/4 cell lines tested (D19 and D34), to a much lesser degree in one (D20), and not at all in another (D38: Figure 2A). D38 is unusual in as much as it is immortal but has retained wild type p53 [16]. It also has retained the ability to increase RARβ expression on treatment with ATRA (Figure 3). In most of these iPMOL cell cultures, the lack of RARβ expression is accompanied by methylation of CpG islands in the RARβ promoter (Figure 2). The modulation of RARβ expression by this mean appears to be an initial, reversible effect in oral cancer development, as tissues from later in the process show loss of expression by non-reversible means, such as chromosome 3p deletion [41]. Mortal oral precancer cells retain RARβ2 expression (D6 and D30: Figure 1A), and this is constitutive, rather than only upon RA exposure, which would normally be the case in normal oral epithelial cells [14]. In D20, the dynamic range of the increase in RARβ2 expression is much less than in D19 and D34. Furthermore, treatment with 5-AZA-CdR does not result in an increase in unmethylated CpG islands on the RARβ promoter; thus, the effects seen may be unrelated to any effect on this promoter.

Expression of the retinoid-binding protein cRBP1 is also increased in D20 and D34 on treatment with 5-AZA-CdR, with no additional effect of ATRA treatment (Appendix A). No change in expression was noted in D38. Parallel changes in cRBP1 expression may be required for retinoid function. 

In cells that re-express RARβ2, treatment with ATRA and 5-AZA-CdR in combination results in accumulation of cells in G2 (Figure 5) and increases in the senescence markers SAB-Gal and HIRA (Figure 4). The cell cycle effects seen are maximal on treatment with 5-Aza-CdR alone, and the addition of ATRA results in little additional change. This maintains the known effects of 5-AZA-CdR treatment on the cell cycle, which initiates a G2 arrest [42]. As retinoids induce cell cycle arrest in G1 [43], it appears that the effects of these treatment schedules on the cell cycle are dominated by the overall effect of 5-AZA-CdR. This somewhat calls into question the additional benefit of ATRA treatment. However, in relation to senescence, the addition of ATRA confers an additional increase in the proportion of cells exhibiting senescence markers in D19 and less so in D20. Additionally, treatment with 5-AZA-CdR increases the proportion of cells with markers of senescence in D34, as has been previously reported [16]. This is not further increased by the addition to ATRA. There is very little effect on the cell cycle or on senescence markers in D38. 

The effects seen on the expression of p16^ink4a^ and p21 support the cell cycle effects seen. In D19 and D34, in which the most marked effects on the cell cycle are seen, treatment with 5-AZA-CdR results in expression of p16^ink4a^ and p21, whilst no effects were seen in D20. Conversely, D38, which has wild-type p53, shows strong induction of p21 on 5-AZA-CdR and combined 5-AZA-CdR +ATRA treatment (Figure 5A), but no consistent effects on ether the cell cycle or induction of senescence. Loss of p16^ink4a^ expression, initially by promoter methylation, has been demonstrated as an early event in OSCC development and has been described in PMOLs [44,45]. Thus, some of the effects seen in the cell cycle and senescence may be due to the re-expression of p16^ink4a^ mediated by 5-AZA-CdR. However, p16 mediated pro-senescence effects have long been associated with arrest in G1, which has not been demonstrated in this study. More recent investigations have shown that the senescence program can be initiated in G2 and that p21 is an important mediator of this process [46]. Whilst it is possible that some of the effects on the expression of p21 indicate that senescence is also being induced via p21 at this point in the cell cycle (in addition to the direct 5-AZA-CdR-mediated accumulation of cells in G2), the lack of effects seen in D38, which has retained WTp53, does not support this conclusion. 

Overall, the results presented demonstrate the variability in response to these agents in cell culture. Response to treatment (defined as an increase in senescence markers and/or alteration in the cell cycle) is variably related to the ability to re-express RARβ2, cRBP1, and p16^ink4a^ in iPMOL cells, but the majority of the effect is seen upon treatment with 5-Aza-CdR alone, with only modest additional effects on treatment with ATRA, even in circumstances where RARβ is re-expressed. Assessment of all these factors would be needed in patient samples to predict the response to therapy with retinoid and de-methylation agents. Furthermore, it is true that assessment of these effects in monolayer cell culture is sub-optimal for assessment of effects (particularly in the assessment of effects on differentiation and the potential effects of the stroma on retinoid metabolism) and use of a more robust 3D TEM model may be required [47]. Our initial investigations of this have found that 3D mucosal models produced using our standard procedures are insufficiently robust to withstand 5-Aza-CdR treatment (Figure 3C).

However, the effects on the D34 effect suggests that a beneficial effect may be possible, as has been previously suggested [16]. As most of the effects seen in this cell culture system are mediated by 5-Aza-CdR, it is unclear how much extra benefit is gained by the addition of ATRA. Additionally, increased efficacy has been demonstrated in other 2D and 3D cell culture systems by the use of newer synthetic retinoids [48], and investigation of the effects of these may also demonstrate increased effects in oral cells. 

## 5. Conclusions

In conclusion, RARβ is constitutively expressed in the mortal dysplastic cells, which undergo replicative senescence. Loss of RARβ by promoter methylation, along with the inactivation of p16 and activation of telomerase, is associated with immortalisation in dysplastic keratinocytes. Treatment of immortal dysplastic cells with 5-AZA-CdR in combination with ATRA leads to the re-expression of RARβ (and in some cases, p16), with a resultant increase in senescence. The explanation for the repression of RARβ expression in D19, D20, and D34 was promoter hypermethylation, emphasising epigenetic mechanisms in oral cancer pathogenesis, and the importance of the combination of drugs for chemoprevention. However, defining the extent to which ATRA per se contributes to these effects is difficult, given the wider effects of 5-AZA-CdR in these cells.

## Figures and Tables

**Figure 1 cancers-13-04064-f001:**
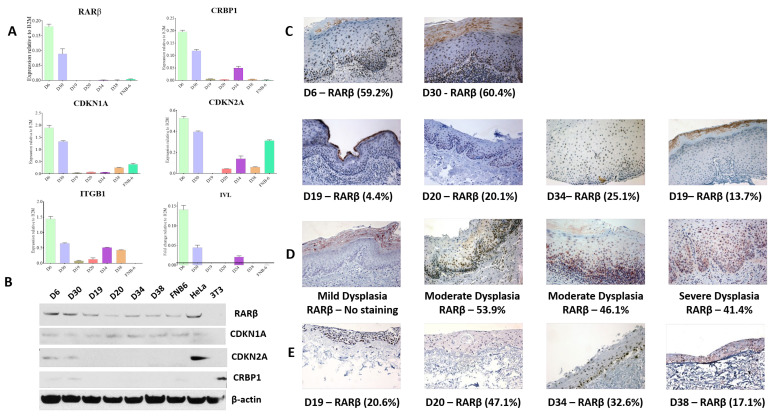
Baseline expression of RARβ2 mRNA and total RARβ protein levels in PMOL cell lines and tissues. (**A**) Assessment of baseline RARβ2 mRNA expression by qPCR demonstrated significantly higher expression in mPMOL cells (D6 and D30) compared to the iPMOL cells and FNB6TERT cells (*p* < 0.03). The mRNA expression of cell cycle regulators, CDKN1A and CDKN2A, was also higher in the mPMOL cells when compared to iPMOL cells (*p* < 0.002) and higher expression of cRBP1 and differentiation markers ITGB1 and IVL was also observed (*p* < 0.05, Kruskal–Wallis test). (**B**) Western blotting of total RARβ, CDKN1A, and CDKN2A with β-actin as a loading control. HeLa cells and i3T3 cells were added as experimental controls. These show a similar pattern of expression to that seen in the qPCR, albeit the Western blot shows total RARβ, rather than RARβ2. (**C**) Immunohistochemical expression of total RARβ in the FFPE biopsy tissues corresponding D6, D30, D19, D20, D34, and D38 showed lower RARβ expression in tissues from which iPMOLs were derived compared to those from which mPMOLs were generated. (**D**) Immunohistochemical expression of RARβ in the extended clinical cohort (*n* = 10) demonstrated variable proportions of total RARβ expression, which was not related to the grade of epithelial dysplasia. (**E**) Immunohistochemical expression of RARβ in 3D TEM models constructed using iPMOL cells showed variable staining for RARβ, but in every case, higher than in the matched biopsy material (Figure 2C).

**Figure 2 cancers-13-04064-f002:**
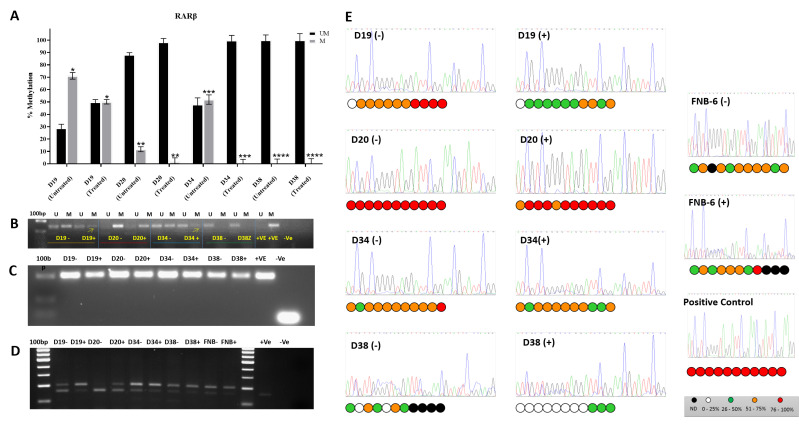
Promoter methylation analysis of the RARβ gene. (**A**) The proportion of RARβ-promoter methylation in D19, D20, D34, and D38 iPMOL cell lines, without (untreated) and treated with 5-AZA-CdR, as detected using the EpiTect Methyl II PCR Assay kit. The % of promoter methylation in D19, D20 and D34 reduced on treatment with 5-AZA-CdR (* *p* < 0.0001, ** *p* < 0.0017, *** *p* < 0.0001 **** p< 0.00005), whereas in D38, the promoter was completely unmethylated prior to treatment. (**B**) Using methylation-specific PCR on DNA from 5-AZA-CdR untreated and treated cells, the RARβ-promoter region (146 bp, consisting of 14 CG sites), was amplified using primers specific for methylation (M) and unmethylation (U), and the products run on a 3% agarose gel. In D19, D20, and D34 cell lines, there was decrease in the intensity of the methylated band’s intensity after treatment with 5-AZA-CdR. (**C**) Using bisulfite-specific primers in both treated (+) and untreated (−) cell lines, a 207 bp promoter region was amplified by PCR. Bands in the gel represent this product which was subsequently treated with TaiI restriction enzyme. (**D**) Treatment of the 207 bp product with TaiI restriction enzyme demonstrates differing sizes of methylated (larger fragment; upper band) and unmethylated (smaller fragment: lower band) fragments. The proportion of methylated and unmethylated fragments are altered in D19 and D34, and there is a reduction in the methylated fragment in D20. The unmethylated RARβ-promoter region in D38 remained unchanged before and after treatment. This pattern of alteration is in keeping with the EpiTect Methyl II PCR and Methylation-specific PCR data. (**E**) Sequencing of the bisulfite-treated DNA 207 bp product allowed calculation of the percentage of methylation at specific sites in the promoter (11 sites). The figure shows representative electropherograms of bisulfite sequencing reads from all iPMOL cells and FNB6TERT. From these, the percentage of CG methylation was calculated by comparing the peak height of cytosine with the sum of the heights of cytosine and thymine signals as represented. Abbreviation. UM = unmethylated fraction of DNA; M = methylated fraction of DNA; (+) treated; (−) untreated.

**Figure 3 cancers-13-04064-f003:**
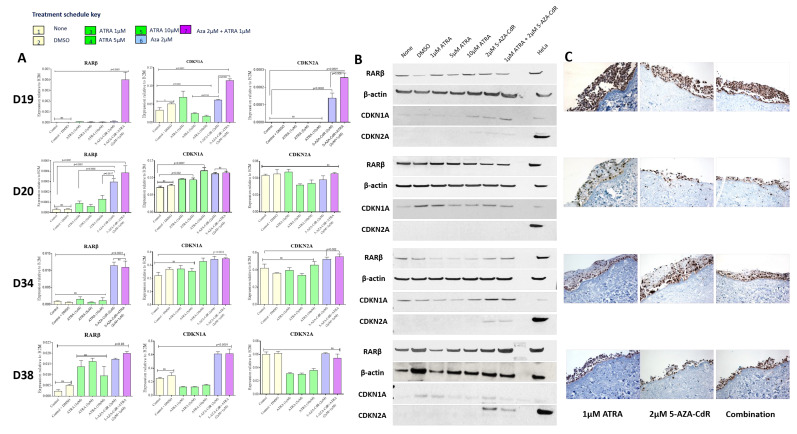
Expression of RARβ and cell cycle markers in iPMOL cells when treated with varying concentrations of ATRA and 5-AZA-CdR used either alone or in combination. (**A**) Expression RARβ2, CDKN1A, and CDKN2A was assessed by qPCR in D19, D20, D34, and D38 cells after treatment with varying doses of ATRA, 5-AZA-CdR alone, or a combination of 5-AZA-CdR and ATRA. Increased expression of RARβ was noted in D19, D20, and D34 (*p* < 0.0001) when treated with a combination of 5-AZA-CdR and ATRA or 5-AZA-CdR alone. A concomitant increase in expression of CDKN1A and CDKN2A was observed in D19 and D38 (*p* < 0.002). D38 showed an increase in the CDKN1A expression proportionate to the re-expression of RARβ on treatment with ATRA and a combination of 5-AZA-CdR and ATRA (*p* < 0.0001). Repeated-measures one-way ANOVA with post hoc Tukey’s test was performed to determine the statistical significance (*p* < 0.05). M = methylated; UM = unmethylated. (**B**) Western blotting for RARβ, CDKN1A, and CDKN2A upon treatment with the varying concentrations of ATRA and 5-AZA-CdR or in combination confirmed their re-expression in the iPMOL cells. M = methylated; U = unmethylated. (**C**) Expression of RARβ was assessed in 3D TEMs constructed using each of the iPMOL cells by immunohistochemistry. The matched untreated TEMs are shown in Figure 1E. Total RARβ staining of the untreated and treated cells with ATRA showed a significant difference in 3D tissue-engineered oral mucosal models before and after treatment. The models were insufficiently robust to withstand the combination of ATRA and 5-AZA-CdR.

**Figure 4 cancers-13-04064-f004:**
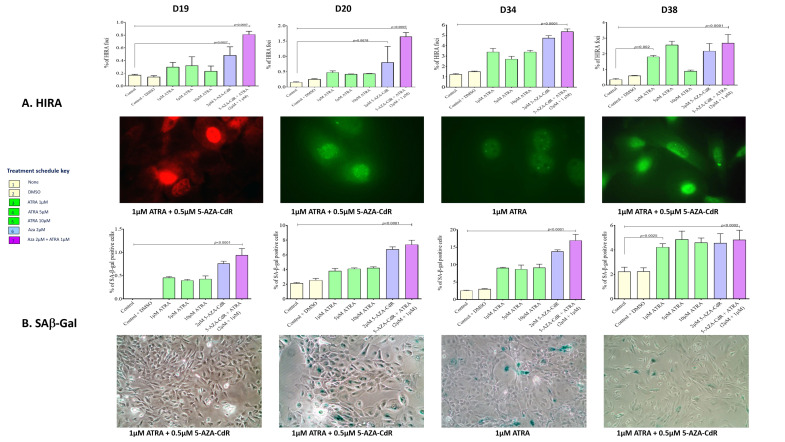
Senescence in ATRA and/or 5-AZA-CdR treated iPMOLs was assessed by assessment of SAH foci and SA-β-gal activity assay. (**A**) Assessment of senescence-associated heterochromatic foci (SAHF) was undertaken using immunofluorescent assessment of HIRA. The mean frequency of HIRA foci showed a higher accumulation in iPMOL cells (D19, D20, D34, and D38) when treated with the combination of 5-AZA-CdR with ATRA or alone compared to the control (*p* < 0.0007). (**B**) Mean percentage values of SA-β-gal positive cells were significantly higher in D20 and D34 cell lines (*p* < 0.0001) than in D19 and D38 when compared to the untreated controls. D34 and D38 cells showed significant increase in the HIRA foci and SA-β-gal activity even when treated alone with the various concentrations of ATRA and/or 5-AZA-CdR. Statistical significance was determined using one-way ANOVA with post hoc Dunnett’s test (*p* < 0.05).

**Figure 5 cancers-13-04064-f005:**
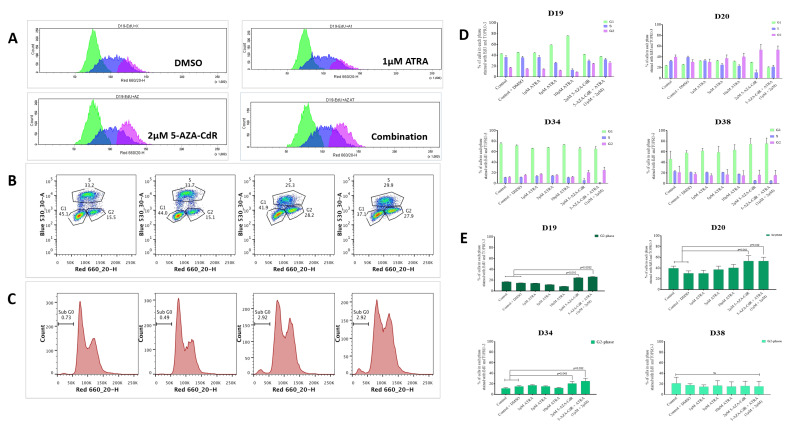
Cell cycle analysis and the sub-GO content by flow cytometry. (**A**) A representative image of cell cycle analysis showing G1-S-G2M peaks assessed by flow cytometry with EdU in ATRA and 5-AZA-CdR treated D19 cells. (**B**) A representative flow cytometry dot plot of EdU-labelled D19 following treatment with ATRA, 5-AZA-CdR, and a combination of 5-AZA-CdR with ATRA showing a slowdown of cell turnover and G2-M arrest. (**C**) Representative histogram of treated D19 cell line highlighting sub-G0 content of fragmented DNA from the apoptotic cells. (**D**) The bar charts represent a summary of the proportions of cells in cell cycle phases G1, S, and G2 in D19, D20, D34, and D38 under various treatment conditions. (**E**) The bar chart represents the cell turnover with ATRA and a block in G2/M phase transition with 5-AZA-CdR and ATRA. An increase in G2-M phase cell accumulation was noted in the D19, D20, and D34 cell lines (*p* < 0.05), whereas no significant difference was noted in the D38 cell line by one-way ANOVA with post hoc Tukey’s test.

**Table 1 cancers-13-04064-t001:** Details of the PMOL cell lines used in the study with original diagnosis and expression of RARβ in the corresponding parent tissues.

Cell Type	Site	Clinical Type	Dysplasia Grade	Phenotype	RARβ Expression (%)
D6	Posterior tongue	Leukoplakia	Moderate	Mortal (25PD)	59.2
D30	Floor of the mouth	Leukoplakia	Mild	Mortal (30PD)	60.0
D19	Lateral tongue	Erythroplakia	CIS	Immortal	4.4
D20	Lateral tongue	Leukoplakia	Moderate	Immortal	20.1
D34	Posterolateral tongue	Leukoplakia	Moderate	Immortal	25.1
D38	Lateral tongue	Leukoplakia	Mild	Immortal	13.7

**Table 2 cancers-13-04064-t002:** Details of the extended OED patient cohort.

No	Sex	Age	Site	Epithelial Dysplasia	Mean RARβ%
P1	F	75	Mandibular Gingiva	Mild	51.2
P2	M	71	Ventral Tongue	Moderate	57.0
P3	M	36	Floor of mouth	Severe	44.9
P4	M	60	Soft Palate	Mild	53.2
P5	F	57	Floor of mouth	Moderate	32.8
P6	M	73	Lateral Tongue	Mild	NED
P7	M	57	Ventral Tongue	Mild	60.9
P8	M	62	Lateral Tongue	Moderate	46.1
P9	M	42	Lateral Tongue	Severe	41.0
P10	M	41	Ventral Tongue	Moderate	43.1

NED: no expression detected.

**Table 3 cancers-13-04064-t003:** RARβ expression in 3D-tissue-engineered oral mucosa models before and after treatment with 1 μM ATRA.

3D Models	RARβ % before Treatment	RARβ % after Treatment with 1 μM ATRA
D19	20.7	66.2
D20	47.1	54.6
D34	32.56	44.3
D38	17.1	39.3

## Data Availability

The data presented in this study are available on request from the corresponding author.

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
