# Peer review of "RARβ Expression in Keratinocytes from Potentially Malignant Oral Lesions: The Functional Consequences of Re-Expression by De-Methylating Agents"

_cancers, 2021, doi:10.3390/cancers13164064_

Round 1

Reviewer 1 Report

In line 85 the authors state"studies focussed " and they cite just one article.If you mention "studies" please cite more articles because is in plural.

At the end of the chapter introduction the authors have to mention the aim of this article and what was the null hypothesis.

In line 545 the authors speak about Figure 1 and cite article 14 after that they mention that previous work was extended in this report. Do the authors have the right to publish this image?

In line 566 the authors mention 'Previous studies" and they cite just one study. Please report to more studies.

The conclusions in my opinion should be moved in the discussion and please insert some pertinent conclusions of this study.

The conclusions should be just short concise, with one or two paragraphs to end the main text.

Author Response

Thank you for your comments on the paper, which have been very constructive and helpful.  Please find a response to each point below:

Comment 1: In line 85 the authors state "studies focussed" and they cite just one article. If you mention "studies" please cite more articles because is in the plural.

Response 1: We have added some other studies that focus on the identification of biomarkers, which may predict retinoid sensitivity [1–3].

Comment 2: At the end of the chapter introduction the authors have to mention the aim of this article and what was the null hypothesis.

Response 2: Thank you for this comment, as t will focus the start of the paper. We have included a short note mentioning the aim of the study and state the null hypothesis at the end.

“Silencing of RARβ occurring as an early event in head and neck carcinogenesis, due to DNA promoter hypermethylation could be reversed by a demethylating agent, 5-Aza-2’-deoxycytidine (5-AZA-CdR) [4,5]. In the present study, we aim to confirm the de novo methylation of RARβ in oral epithelial dysplasia and understand its role in cellular immortalization and abrogation of cellular senescence. We test the hypothesis if administration of 5-AZA-CdR alone and/or in combination with All trans-retinoic acid (ATRA) leads to reexpression of RARβ, reversal of immortalisation, and reinduction of the senescence programme in a panel of immortalized primary cultures”.

Comment 3: In line 545 the authors speak about Figure 1 and cite article 14 after that they mention that previous work was extended in this report. Do the authors have the right to publish this image?

Response 3: We confirm that all the images in the manuscript are from our work. We were able to substantiate the previous work of McGregor et al. in our study, but the experimentation was different, and no figures or data have been reused.

Comment 4: In line 566 the authors mention 'Previous studies" and they cite just one study. Please report to more studies.

Response 4: We have included another related study (Chakravarti, et al) where there was a significant decrease in RARbeta immunopositivity in hyperplastic lesions compared with normal oral mucosa as well as in OSCC compared with normal tissue. However, inconsistency in staining has been attributable to the availability of antibodies against specific isoforms, which may sometimes limit the scope of correlating the qPCR results with the Western blots.

Comment 5: The conclusions in my opinion should be moved in the discussion and please insert some pertinent conclusions of this study. The conclusions should be just short concise, with one or two paragraphs to end the main text

Response 5: We have moved the current conclusion back into the discussion section of the manuscript and have inserted a concise paragraph highlighting the pertinent findings in a revised conclusion.

“In conclusion, RARb is constitutively expressed in the mortal dysplastic cells which undergo replicative senescence. Loss of RARb by promoter methylation, along with the inactivation of p16 and activation of telomerase is associated with immortalization in dysplastic keratinocytes. Treatment of immortal dysplastic cells with 5-AZA-CdR in combination with ATRA leads to re-expression of RARb (and in some cases, p16), with a resultant increase in senescence. The explanation for the repression of  RARb expression in D19, D20 and D34 was promoter hypermethylation, emphasizing epigenetic mechanisms in oral cancer pathogenesis and the importance of the combination of drugs for chemoprevention. However, defining the extent to which ATRA contributes to these effects is difficult, given the wider effects of 5-AZA-CdR in these cells.”

Reviewer 2 Report

cancers-1305440: Raghu Radhakrishnan et al.

Importantly: The figures provided in the PDF document are incomplete, not allowing a full analysis of the work.

The manuscript entitled “RARB expression in keratinocytes from potentially malignant oral lesions: the functional consequences of re-expression by de-methylating agents.” attempts to address new issues regarding the use of retinoids as chemopreventive agents to treat oral malignant oral lesions. Among the three-retinoic acid receptors, RARβ is frequently deleted or epigenetically silenced at early stages in tumor progression and there is compelling evidence that RARβ may correspond to a tumor suppressor. In this work, authors use several immortal potentially malignant oral lesions (PMOL) cell cultures, as well PMOL tissues, to investigate the effects of a demethylating agent and of all trans retinoic acid on the isoform RARβ2 expression, the methylation status of the RARβ promoter, the growth inhibition, and the senescence.

The title and abstract are informative and give a clear idea of what to expect from the paper.

Introduction and discussion sections are adequately documented.

Regarding the experimental part, a range of relevant and conventional approaches are used and described in sufficient detail.

Even if this article represents a substantial amount of work, the scientific scope of this article is limited. The document is tedious to read, and the results correspond to observations that are difficult to interpret and do not allow solid conclusions to be drawn.

Also the figures provided in the PDF document are incomplete, not allowing a full analysis of the work.

In addition, without being exhaustive, the following is a list of few points that should be clarified and/or improved

Introduction:

1/ Authors use a trivial name for RARβ. A logical numbering system and receptor code, supporting the trivial names, was made by the International Committee of Pharmacology Committee on Receptor Nomenclature and Classification (NC-IUPHAR). In each manuscript dealing with nuclear receptors, it is recommended that the receptors be identified by the official names at least once in the summary and the introduction. Once the name has been established, authors may use the trivial name for the remainder of the manuscript.  For instance, the trivial names and the formal nomenclature for RARβ is NR1B2.

2/ Abbreviations RARβ and RARβ2 are used. Please write at least once “retinoic acid receptor”.

3/ For readers not familiar with the retinoic acid receptors, please explain what is meant by RARβ2 (isoform of RARβ) and that RARβ is a subtype of RAR.

Materials end methods

4/ 2.6 and 2.7 (Promoter Methylation Analysis) are identical

Results

5/ Figures are incomplete and the text is often too small

6/ Figure 2E: although it is possible to deduce the meaning of the colors, please specify the color code

7/ Figure 3A: contrary to what is written, I do not observe a significant increase in RARβ for D19 with AZA alone.

8/ Figure 3A: Authors state that the results imply that the response is cell line-specific and dose-dependent. Variable variations in the expression of the proteins studied are observed with the different concentrations of ATRA used (1, 5, and 10µM), which do not allow a dose-response relationship to be established. This result is curious and uninterpretable. Moreover, why are such high doses of ATRA used? At these concentrations independent effects of RARs may occur (the Kd of ATRA for RARβ is of the order of 10 nM).

Author Response

Thank you for your helpful and constructive comments on the manuscript.  We have replied to each in turn, as outlined below:

Reply to initial comments:  We wish to apologize for the incomplete figures and this has now been rectified as outlined below.  We acknowledge that the experimentation covers a wider range of techniques, but has not resulted in a clear set of conclusions.  This means that there are many issues to discuss regarding the possible explanations for this. Whilst the lack of a clear conclusion has been disappointing for us, we thought it useful to present the data as it may explain some of the inconclusive results from the initially very promising retinoid clinical trials and may allow for the development of other paths forward in the clinic.

Revisions in Introduction:

Comment 1: Authors use a trivial name for RARβ. A logical numbering system and receptor code, supporting the trivial names, was made by the International Committee of Pharmacology Committee on Receptor Nomenclature and Classification (NC-IUPHAR). In each manuscript dealing with nuclear receptors, it is recommended that the receptors be identified by the official names at least once in the summary and the introduction. Once the name has been established, authors may use the trivial name for the remainder of the manuscript. For instance, the trivial names and the formal nomenclature for RARβ is NR1B2.

Response 1: We concur with the views of the reviewer: we should have identified the receptor by its official name. However, we have referred to the HUGO Gene Nomenclature Committee for the approved name and approved symbol of retinoic acid receptor-beta, which is RARβ -HGNC:9865. We have also added the receptor code NR1B2 as recommended by the reviewer in the introductory section.

Comment 2: Abbreviations RARβ and RARβ2 are used. Please write at least once “retinoic acid receptor”.

Response 2: We have expanded the abbreviation RARβ when we used it in the first instance in the revised manuscript.  

Comment 3: For readers not familiar with the retinoic acid receptors, please explain what is meant by RARβ2 (isoform of RARβ) and that RARβ is a subtype of RAR.

Response 3: We have inserted text into the introduction regarding the the RARβ splice variants and a specific subtype of retinoic acid receptor in the revised manuscript

Revision in materials and methods:

Comment 4: 2.6 and 2.7 (Promoter Methylation Analysis) are identical

Response 4: We have now corrected this error in proofing the manuscript: we have removed paragraph 2.7 and renumbered the methods section in the revised version of the manuscript.

Revisions in Results:

Comment 5: Figures are incomplete and the text is often too small

Response 5: We have noted that all of the figures have been cropped on the right edge.  We apologize for this, which has occurred in moving the figures into the main text.  These were provided in landscape orientation, but have been inserted (by the MDPI editorial team) in portrait mode.  This has now been rectified. The text has also been enlarged and also the images resized for improved resolution.

Comment 6: Figure 2E: although it is possible to deduce the meaning of the colors, please specify the color code

Response 6: Figure 2E was provided as a portion of the image on the right side was cropped – which contained the colour key. We apologize that this was missing. An additional key for the treatment schedule has also been added to Figure 3 and Figure 4, for clarity.

Comment 7: Figure 3A: contrary to what is written, I do not observe a significant increase in RARβ for D19 with AZA alone.

Response 7: The expression of RARβ, when treated with 5-AZA-CdR alone, is significant (0.000115) when compared with the control (0.000021) which is ~5 fold higher. However, the presence of 0.004010 expression units in the same graph when cells were treated in combination (approx. ~190 fold), means that in the scale bar required, this difference is not evident. They are marginally significant albeit not high as when treated with a combination of 5-AZA-CdR with ATRA. This is in keeping with our WB data, in which this change in expression is evident.

Comment 8: Figure 3A: Authors state that the results imply that the response is cell line-specific and dose-dependent. Variable variations in the expression of the proteins studied are observed with the different concentrations of ATRA used (1, 5, and 10μM), which do not allow a dose-response relationship to be established. This result is curious and uninterpretable. Moreover, why are such high doses of ATRA used? At these concentrations, independent effects of RARs may occur (the Kd of ATRA for RARβ is of the order of 10 nM).

Response 8: We thank the reviewer for this insightful comment. It is known that ATRA, when used in the order of 10nM concentration, has a specific ligand-mediated receptor function (Germain et al, Pharmacol. Rev. 2006, 58, 760–772). However, in this project, we were trying to assess the effective dosage for re-expression of RARb to use it in combination with 5-AZA-CdR.  The doses we used are similar to those used in the combined treatment of oral cells with AZA and ATRA reported in references 16 and 17. Furthermore, the dose is within the range of ATRA dose used in reference 20, with very little difference in effect reported between 10-6M and 10-8M, but some inhibition of growth was seen using 10-5M. Using these previously reported doses in our cells, ATRA contributed to a modest response on the reexpression of genes (Figure 3), and we have discussed the role ATRA is playing in addition to 5-AZA-CdR in the discussion section of the paper. Admittedly there is variation in the expression on treatment with ATRA in cells without the addition of 5-AZA-CdR, but in most cases, there is a marked difference once that was added.  On reflection, repeating the ATRA treatment at various doses after 5-AZA-CdR treatment would have been useful but we did not do this.

Round 2

Reviewer 2 Report

I thank the authors for this revised version.